# Vaccination Recommendations in Solid Organ Transplant Adult Candidates and Recipients

**DOI:** 10.3390/vaccines11101611

**Published:** 2023-10-18

**Authors:** Mauro Viganò, Marta Beretta, Marta Lepore, Raffaele Abete, Simone Vasilij Benatti, Maria Vittoria Grassini, Stefania Camagni, Greta Chiodini, Simone Vargiu, Claudia Vittori, Marco Iachini, Amedeo Terzi, Flavia Neri, Domenico Pinelli, Valeria Casotti, Fabiano Di Marco, Piero Ruggenenti, Marco Rizzi, Michele Colledan, Stefano Fagiuoli

**Affiliations:** 1Gastroenterology Hepatology and Transplantation Unit, ASST Papa Giovanni XXIII, 24127 Bergamo, Italysfagiuoli@asst-pg23.it (S.F.); 2Pulmonary Medicine Unit, ASST Papa Giovanni XXIII, 24127 Bergamo, Italy; m.beretta@asst-pg23.it (M.B.); fdimarco@asst-pg23.it (F.D.M.); 3Unit of Nephrology and Dialysis, ASST Papa Giovanni XXIII, 24127 Bergamo, Italy; mlepore@asst-pg23.it (M.L.); pruggenenti@asst-pg23.it (P.R.); 4Cardiology Division, ASST Papa Giovanni XXIII, 24127 Bergamo, Italycvittori@asst-pg23.it (C.V.); 5Infectious Diseases Unit, ASST Papa Giovanni XXII, 24127 Bergamo, Italy; sbenatti@asst-pg23.it (S.V.B.); mrizzi@asst-pg23.it (M.R.); 6Section of Gastroenterology & Hepatology, Department of Health Promotion Sciences Maternal and Infant Care, Internal Medicine and Medical Specialties, PROMISE, University of Palermo, 90128 Palermo, Italy; 7Department of Organ Failure and Transplantation, ASST Papa Giovanni XXIII, 24127 Bergamo, Italy; scamagni@asst-pg23.it (S.C.); fneri@asst-pg23.it (F.N.); dpinelli@asst-pg23.it (D.P.); mcolledan@asst-pg23.it (M.C.); 8Cardiothoracic Department, ASST Papa Giovanni XXII, 24127 Bergamo, Italy; aterzi@asst-pg23.it; 9Pediatric Hepatology, Gastroenterology and Transplantation Unit, ASST Papa Giovanni XXII, 24127 Bergamo, Italy; vcasotti@asst-pg23.it; 10Department of Health Sciences, University of Milan, 20158 Milan, Italy; 11Department of Renal Medicine, Clinical Research Centre for Rare Diseases “Aldo e Cele Daccò”, Institute of Pharmacologic Research “Mario Negri IRCCS”, Ranica, 24020 Bergamo, Italy; 12Department of Medicine, University of Milan Bicocca, 20126 Milan, Italy

**Keywords:** solid organ transplantation, vaccination, chronic diseases, liver, lungs, heart, kidney, infections, prevention

## Abstract

Prevention of infections is crucial in solid organ transplant (SOT) candidates and recipients. These patients are exposed to an increased infectious risk due to previous organ insufficiency and to pharmacologic immunosuppression. Besides infectious-related morbidity and mortality, this vulnerable group of patients is also exposed to the risk of acute decompensation and organ rejection or failure in the pre- and post-transplant period, respectively, since antimicrobial treatments are less effective than in the immunocompetent patients. Vaccination represents a major preventive measure against specific infectious risks in this population but as responses to vaccines are reduced, especially in the early post-transplant period or after treatment for rejection, an optimal vaccination status should be obtained prior to transplantation whenever possible. This review reports the currently available data on the indications and protocols of vaccination in SOT adult candidates and recipients.

## 1. Introduction

Immunosuppression is the major predisposing factor for the development of infections, influencing both their incidence and severity. The immunodepressed population includes not only patients treated with immunosuppressive (IS) drugs for solid organ transplant (SOT), but also individuals with end-stage diseases. These patients bear a remarkable risk of infections due to immune system dysfunction and dysregulation of innate and adaptive immunity [1,2,3,4,5,6,7]. In particular, in patients with kidney and liver failure, expression of toll-like receptors (TLR) and B and T lymphocyte proliferation may be decreased, response to antigenic stimuli is impaired, capacity of phagocytosis is limited and apoptosis is increased. Moreover, there is an impairment of leukocyte and endothelial function and a low-grade inflammation with overproduction of inflammatory cytokines inducing oxidative stress. Additional contributors to the risk of infection in all these SOT candidates could be the presence of IS-related co-morbidities and diabetes mellitus.

Among the different approaches for preventing infections, vaccines are paramount, especially in the perspective of a future SOT, when IS will boost the infectious risk and further limit the immune system responsiveness. On the other hand, due to the compromised immune response, serological response to vaccines in the end-stage of most chronic diseases may not be as optimal as in healthy controls [8,9]. As vaccines demonstrate superior immunogenicity when given earlier in the course of any inflammatory or chronic disease, even if the definition of most of the schedules is still uncertain, the best strategy is to verify as soon as possible the vaccination status and the response to previous vaccines in all patients with advanced organ disease to resolve any possible immunity gaps.

The immunization of SOT patients presents several concerns due to the following reasons:Vaccination schedules generally rely on few data without trials specifically designed for this issue;The timing of immunization may be challenged by the dynamics of the immunosuppressive regimens;Live attenuated vaccines (LAVs) are generally avoided in SOT recipients whereas inactivated vaccines have proven to be safe and not significantly associated with rejection episodes;Vaccines tolerance is poorly known in end-stage chronic diseases.

Although LAVs are generally to be avoided in recipients of SOT, this view has been recently questioned. Several studies were performed in paediatric patients using the measles mumps and rubella (MMR) vaccine and varicella zoster vaccine (VZV) after SOT. However, these studies evaluated different timing, patterns of sero-conversions and needs for booster doses. Overall, the evidence was that live vaccines are safe in selected SOT patients with minor adverse effects, similar as in healthy children, and with a good development of seroprotection, but with the need of repeated immunizations to ensure a sustained response. Moreover, no cases of infection with the attenuated vaccine strain were reported [10,11,12,13,14]. A consensus was held in 2018 leading to specific indications for the administration of MMR and VV vaccines after SOT [15]. Their indication is to address such intervention only to the cases where all precautions have been adopted, or otherwise, in cases where the risk of vaccination is expectedly lower than the opposite (i.e., during an infectious outbreak or before an unavoidable travel to endemic risk areas). In general, these authors recommend limiting live MMR or VZV vaccines after SOT (avoiding the combination of the MRR and VZV vaccine) to the following:-Patients who are clinically well, more than one year after liver or kidney transplant and two months after acute rejection episodes;-Patients who can be closely monitored after vaccination;-Patients under “minimum immune suppression and “minimum immune criteria. The “minimum immune criteria” are defined as absolute lymphocyte count >1500 or >1000 cells/μL in children ≤6 and >6 years, respectively; CD4 >700 or >500 cells/μL in children ≤6 and >6 years, respectively; normal IgG levels; ability to produce protective antibodies to inactivated vaccines prior to the administration of live viral vaccines. The “minimum immune suppression” is defined as steroid <2 mg/kg/day or cumulative dosage <20 mg/day, tacrolimus and cyclosporine serum levels <8 ng/mL and <100 ng/mL, respectively. However, post-transplant live vaccines are still not offered by all centres, mainly because of concerns about safety and lack of efficacy data.

Notably, the risk of live vaccines in SOT recipients is linked to vaccine-mediated disease. Indeed, the use of Yellow Fever Vaccine has been associated with acute antibody-mediate renal graft rejection [16].

In some cases, if a corresponding inactivated vaccine is available, this ought to be utilized (i.e., in case of typhoid vaccination). Unfortunately, no data are available on the level of protection warranted against a primary VZV infection by recombinant zoster vaccines. Notwithstanding the foregoing, all efforts should be made to vaccinate the family members and closest relatives of immunosuppressed patients with LAVs (principally MMR and VZV).

The aim of this review is to assess all the main indications to active immunization in adult patients with advanced heart, kidney, liver and lung disease and in SOT recipients, evaluating the most relevant vaccination recommendations in this setting for the population. Since intestinal and pancreas transplant are usually performed in a multi-visceral context in combination with the liver and/or the kidney, respectively, we have assimilated these two transplant procedures to liver and kidney recipients, respectively.

## 2. Lung Transplant Candidates

Lung transplant (Lu-T) candidates are at increased risk of infectious complications which may lead to substantial morbidity and mortality. In order to improve survival and quality of life, infections in such patients should be prevented by vaccination (Table 1). The vaccination status must be assessed beforehand and Lu-T candidates should be immunized as early as possible to optimize protection against vaccine-preventable illnesses, because vaccine responses are reduced after transplantation, due to IS therapy. Patients waitlisted for Lu-T should follow the indication for vaccination of the Infectious Diseases Society of America (IDSA) guidelines [17]. In Lu-T candidates and recipients, it is still uncertain whether vaccinations can offer a proper clinical protection as the ability to mount adequate immune responses may be compromised.

### 2.1. Influenza

Influenza is a common infection in SOT recipients and correlates with high morbidity, mortality and risk of rejection. Thus, annual administration of the seasonal inactivated influenza vaccine is recommended for all Lu-T candidates.

### 2.2. Streptococcus Pneumoniae

Immune status assessment before Lu-T revealed that more than half of Lu-T candidates presented serum concentrations of immunoglobulin below the normal range. However, although post-vaccination protective antibody levels increased, the majority of patients did not reach protective serum levels (>1.3 μg/L).

The 2013 IDSA guidelines recommended administering the 13-valent pneumococcal conjugate vaccine (PCV13) and 23-valent pneumococcal polysaccharide vaccine (PPSV23) to Lu-T candidates. In vaccine-naïve patients, PCV13 should be administered first, followed by PPSV23 at least 8 weeks later. A booster of PPSV23 can also be administered after 5 years.

### 2.3. Hepatitis A Virus and Hepatitis B Virus

The hepatitis A virus (HAV) (two-dose series repeated after 6–12 months) and hepatitis B virus (HBV) (three-dose series with the first two doses at a ≥4-week distance, and a third dose after 4–6 months) inactivated vaccines can be administered before Lu-T. Unvaccinated HBV adults who are immunocompromised should receive two doses of Engerix-B 20 mcg/mL administered concomitantly in a four-dose schedule at 0, 1, 2 and 6 months. Unvaccinated or not fully covered patients should receive the missing doses to complete the three-dose series with the second dose 1 month after the first and third dose ≥2 months after the second dose (and ≥4 months after the first). Patients who need both the HBV and HAV vaccines can receive a three-dose schedule of combined HAV and HBV vaccine preparation at 0, 1 and 6 months. The first and second doses should be administered at a ≥4-week distance, and the second and third at a ≥5-month distance. Alternatively, the vaccine can be administered with an accelerated schedule of four doses: on days 0, 7 and between days 21 and 30, followed by a booster at 12 months after the first dose. HAV vaccines are prepared from inactivated virus. Depending on the type of vaccine, adults are given the vaccine in a series of two doses at 0 and 6–12 months (Havrix) or 0 and 6 to 18 months (Vaqta).

### 2.4. SARS-CoV-2

All transplant candidates are eligible for vaccination, unless contraindicated, since these patients are a particularly vulnerable group for coronavirus disease 2019 (COVID-19), the lungs being the primary target for SARS-CoV-2.

### 2.5. Herpes Zoster Virus

The recombinant vaccine for the herpes zoster virus (HZV) is recommended in Lu-T candidates aged ≥50 years. It is administered in two doses, 2 to 6 months apart. It should be completed at least two weeks prior to Lu-T, to help ensure maximal immune response. Transplant candidates <50 years could still be considered for the HZV vaccine, if risk factors are present as a previous episode of herpes zoster or a state of immunodeficiency or immunodepression due to disease or therapy.

### 2.6. Human Papilloma Virus

The quadrivalent HPV vaccine is recommended before Lu-T for all individuals between the ages of 9 and 26 to prevent cervical cancer, anal cancer and anogenital wart. It is administered to transplant candidates in the aforementioned age group who have never received the HPV vaccine or have not completed the three-dose series.

The quadrivalent human papilloma virus (HPV) vaccine prevents cervical cancer, anal cancer and anogenital wart. It is recommended for all Lu-T candidates between the ages of 9 and 26 years in a three-dose series.

### 2.7. Haemophilus Influenzae Type B

*Haemophilus influenzae* type B vaccines help prevent infections by this strain of *Haemophilus*, but not those caused by other strains. It is recommended before Lu-T.

### 2.8. Other Vaccinations

The quadrivalent conjugate meningococcal vaccine (ACYW) and vaccine for serogroup B should be administered in Lu-T candidates in the presence of the following identified risk factors: travellers to high-risk areas, properdin-deficient patients, terminal complement component-deficient patients and those with functional or anatomic asplenia. Recommendations for tetanus, tetanus and diphtheria (Td), or tetanus, diphtheria and pertussis (Tdap) vaccines are the same as those for healthy individuals.

## 3. Lung Transplant Recipients

Several factors might predispose Lu-T recipients to infections, such as previous colonization by multi-drug-resistant microorganisms, contact of the graft with the external environment, suppression of the cough reflex, disruption of the lymphatic drainage system, impaired mucociliary clearance, need for high levels of IS therapy—the highest among SOT recipients—and the fact that in this cohort of patients the antimicrobial therapy is often less effective than in the immunocompetent ones. The planning of the vaccination strategy in transplant recipients must consider that LAVs, i.e., MMR, VZV and intranasal influenza vaccine, are contraindicated after transplantation, because of elevated risk of reactivation. The IDSA recommends waiting at least 4 weeks between LAV administration and subsequent Lu-T, whenever possible. Inactivated vaccines are considered safe in SOTs, and they should be completed at least 2 weeks before Lu-T. Administration of vaccines in the post-transplant period should be resumed not earlier than 6 months after the transplant, when patients are treated with higher doses of IS. A considerable exception is the influenza vaccine, which can be administered in the first month after Lu-T [18].

### 3.1. Influenza

Annual administration of the seasonal inactivated influenza vaccine is recommended for all transplant recipients, starting from the first month following transplantation. Conversely, live attenuated influenza vaccine in the post-transplant period is not recommended because of the high risk of active viral reactivation. Lu-T recipients who have inadvertently received a live formulation of the influenza vaccine or have been exposed to influenza are valid candidates for antiviral prophylaxis. In the post-Lu-T period, randomized studies [19,20] show that protective serum antibody levels are achieved in only about 30% of Lu-T recipients, 4 weeks after influenza vaccination.

Lu-T recipients did not demonstrate any cell-mediate immune response within 4 weeks of vaccination, in terms of levels of IL-2, IL-10, IFN- gamma and granzyme B. Supporting this evidence, a 5-year longitudinal study showed that antibody responses in Lu-T patients who received the influenza vaccine before transplant were higher compared to those in patients who received the vaccine between 13 and 60 months post-transplant (*p* = 0.002). This evidence suggests that patients have a stronger pre-transplant response and consequently that annual seasonal vaccination should begin before transplantation [21]. Furthermore, randomized studies have shown that the administration of high doses or boosters in the same season should be preferred over a single standard dose, since they both warrant greater immunogenicity [22]. In addition to this, trivalent seasonal influenza vaccines have shown variable results among different strains and they may be more immunogenic than others, although the clinical protection is variable. Antibody levels to the three viral antigens included in trivalent vaccines are different between non-transplant (group-control) patients and transplant patients. The humoral immune response to the influenza vaccination was significantly lower in the transplant group for all three viral antigens (A/Sydney, A/Beijing (H1N1), Yamanashi (B)) compared to the control group [22,23].

The effect of prednisone on the response to the influenza vaccination is variable with evidence of normal responses for the monovalent H1N1 vaccine and normal or lower responses for the trivalent vaccine. On the other hand, mTOR inhibitors are known to impair H1N1 antibody titres, whereas very few data are available regarding Basiliximab and azathioprine. Long-lasting data from the literature have shown that azathioprine is not hampering the response to live vaccines when administered at a dose ≤3 mg/kg, while there is conflicting information regarding the recombinant influenza vaccine (Flublok Quadrivalent). A single study has suggested that Basiliximab might enhance the antibody response to the influenza vaccine [23].

### 3.2. Streptococcus Pneumoniae

The 2013 IDSA guidelines recommend administering PCV13 and PPSV23 pneumococcal vaccines to Lu-T recipients. Today, the new 20-valent pneumococcal conjugate vaccine (PCV20) has been approved and recommended for SOT candidates and recipients.

Considering the effects of IS drugs, the available literature confirms that prednisone or dexamethasone, especially when combined with other IS therapies, severely impairs the humoral response and consequently the efficacy of PPSV23 [24,25,26]. In addition, Mycophenolate Mofetil (MMF) has been documented to interact with the immune response in reducing the efficacy of vaccination: it completely disrupts primary and secondary humoral responses to pneumococcal polysaccharide vaccines [27,28].

### 3.3. HAV and HBV

HAV and HBV vaccines can be administered after Lu-T because they are inactivated vaccines. Only scattered data on the HAV vaccination on Lu-T patients are present in the literature [29]. Regarding the HBV vaccine, the existing data show how anti-HB serum levels >10 mIU/L provide adequate protection against the infection [30]. However, the level of anti-HBs in Lu-T recipients declines rapidly after transplantation, probably due to extremely high doses of IS drugs in the setting of Lu-T. Nevertheless, data show that recipients who respond to the vaccine before transplantation tend to maintain a T-cell-mediated “memory” response to HBsAg comparable to that of healthy subjects, despite the rapid decrease in serum antibody titres [31,32]. Corticosteroids, among IS drugs, bear the highest impact on the impaired response to these vaccines, interfering with HBV replication.

### 3.4. SARS-CoV-2

The American Society of Transplantation (AST) recommends vaccinating at least two weeks before Lu-T and indicates that vaccines are unlikely to trigger rejection episodes or to induce severe side effects. In patients who are vaccinated prior to Lu-T or who have been transplanted in the interval between doses, the vaccination or the missing dose should be delayed for at least 1 month after Lu-T and for at least 3 months if T- or B-cell-depleting agents have been used for induction.

Emerging data indicate a poor antibody response to mRNA vaccines in Lu-T patients. Only 25% of 73 Lu-T patients who were administered the two-dose mRNA vaccine had IgG (specific to spike protein) above the defined cut-off [33]. Long-term efficacy is still unknown; therefore, it is recommended that patients be monitored in the long term, and in the future, additional vaccine doses and/or types might be needed. For Lu-T patients, monoclonal antibodies against SARS-CoV-2 are always indicated in case of a pre-exposure prophylaxis.

### 3.5. HZV

The recombinant vaccine for herpes zoster is recommended in Lu-T recipients. It is administered in two doses, 2 to 6 months apart. It should be completed at least two weeks prior to transplantation, to help ensure maximal immune response. All transplant candidates aged ≥50 years should receive the herpes zoster vaccination. Transplant candidates and recipients <50 years could still be considered for the herpes zoster vaccine, if risk factors are present as previous episodes of herpes zoster or a state of immunodeficiency or immunodepression due to disease or therapy.

### 3.6. HPV

The quadrivalent HPV vaccine is recommended in the population between 9 and 26 years to prevent cervical cancer, anal cancer and anogenital wart. It can be administered to Lu-T patients in a three-dose series. The HPV vaccine has proven to be safe and well tolerated in Lu-T recipients [34], but among SOT recipients, it was associated with a lower antibody response in Lu-T patients. Current available data show that high daily IS schedules severely hamper the response to HPV vaccine [35]. Similarly, MMF significantly impairs HPV vaccine efficacy since it lowers HPV 6 and 8 seroconversion rates after 12 months, and is associated with a decline in anti-HPV-16 antibody titres 7 months after Lu-T.

### 3.7. Haemophilus Influenzae Type B

Nowadays, in most countries this vaccine is a routine childhood vaccination, but it is also strongly recommended for immunocompromised adults such as SOT recipients, who bear a peculiar risk of acquiring an invasive form of the infection. Haemophilus influenzae type B vaccines help prevent infections by this strain of Haemophilus, but not those caused by other strains of *Haemophilus influenzae* bacteria. Being a conjugated vaccine, it can be administered in the post-transplant period.

### 3.8. Other Vaccinations

The quadrivalent conjugate meningococcal vaccine (ACYW) and vaccine for serogroup B should be administered in Lu-T recipients only in the presence of risk factors: travellers to high-risk areas, properdin-deficient patients, terminal complement component-deficient patients (including acquired complement deficiency such as prior to starting eculizumab) and those with functional or anatomic asplenia. The efficacy of the meningococcal vaccine is not well studied in Lu-T.

Current indications for tetanus, *tetanus diphtheria* (Td) or *tetanus, diphtheria and acellular pertussis* (TDaP) booster vaccines are the same for healthy individuals and Lu-T patients. However, the indication to repeat the vaccinations every 10 years in Lu-T patients should be revised, since it was observed that in this population antibody concentrations were significantly lower than those in healthy individuals as early as 0–5 years post-vaccination [36]. It is recommended to monitor antibody levels in Lu-T patients at regular intervals to ensure patient safety with adequate antibody seroprotection. IS drugs, such as high-dose prednisone, MMF and m-TOR inhibitor, reduce the efficacy of vaccination. LAVs for measles and mumps are contraindicated after Lu-T. For naïve patients, the last dose of live vaccine should be administered at least 4 weeks before transplantation.

## 4. Heart Transplant Candidates

All heart transplant (HT) candidates should undergo serological testing for the most common viruses: Cytomegalovirus (CMV), Epstein–Barr virus (EBV), VZV, Herpes Simplex Virus (HSV 1–2), HBV, HAV, Human Immunodeficiency Virus (HIV), MMR and SARS-CoV-2, and vaccination, whenever available, is recommended in non-immunized patients according to guidelines [17,37]. Before HT, candidates should be vaccinated with recombinant vaccine against HBV and mRNA vaccines against SARS-CoV-2, influenza, tetanus, HAV, meningococcus and pneumococcus. Effective protection against HBV may allow a broader use of hepatitis core antibody (anti-HBc) positive grafts.

All vaccinations should be completed preferably one month before HT, since during this period, the immune response to the vaccine may vary depending on the patient’s clinical conditions (Table 2).

## 5. Heart Transplant Recipients

After HT, vaccinations should be avoided for the first 3–6 months, during which patients should lead a reduced social and work life [38]. Between the 9th and 12th months post-HT, patients can return to work and wider social engagements. Depending on their medical conditions, patients can then be vaccinated. Their immune response is not predictable, but it is certainly reduced, especially in patients undergoing aggressive IS regimens. After HT, LAVs are not recommended. The use of mRNA technology vaccines is possible, but efficacy may be reduced [39] (Table 2).

### 5.1. Influenza

Vaccinations against influenza are recommended on a yearly basis. Influenza vaccination with inactivated virus is safe and effective without increasing the risk of either rejection episodes or infections because it does not induce allo-sensitization. However, the response in transplant recipients is reported to be lower compared with healthy subjects [40,41,42]. The less expensive influenza vaccines without adjuvants can provide similar efficacy in protecting HT recipients compared to those with adjuvants [43].

### 5.2. Streptococcus Pneumoniae

Pneumococcal vaccination is as safe and effective in HT recipients under IS therapy as in healthy individuals.

### 5.3. HBV

Vaccination against HBV is necessary in HT recipients because they may experience more severe and rapid progression of HBV infection, as well as a reactivation of latent infection under IS treatment [44].

### 5.4. SARS-CoV-2

Recently, Peters et al. reported that SARS-CoV-2 vaccination among 436 HT recipients, was associated with a lower risk of SARS-CoV-2 infection and with fewer complications, hospitalizations and deaths due to COVID-19. Moreover, patients with COVID-19 are at greater risk of severe infection and death compared with immunocompetent individuals [45].

The International Society for Heart and Lung Transplantation (ISHLT) COVID-19 Task Force, advises to delay the SARS-CoV-2 vaccination for at least 1 month in post-HT and at least 3 months from the use of T-cell depleting agents or specific B-cell depletion agents. Vaccination should be delayed for 3 months in patients who received monoclonal antibodies for COVID-19. All the currently available vaccines against SARS-CoV-2 are acceptable in HT recipients [46]. Based on current evidence, a third dose of the mRNA vaccine is recommended for patients who have previously completed a series of two doses of mRNA vaccines. The repeated use of booster vaccines needs to be supported by additional evidence. A meta-analysis conducted by Alhumaid et al. reported only one cellular rejection episode among HT recipients, and showed that the protective benefits of SARS-CoV-2 vaccination far outweighs the risks [47].

### 5.5. Other Vaccinations

Traveling to high-risk destinations is not recommended, but traveling to these areas after the first 2 years post-HT in patients without complications is not contraindicated. Specific travel vaccinations are often based on LAVs which are contraindicated in HT recipients and therefore such patients should receive information on the most appropriate health-related behaviours to be adopted during travel. HT recipients aged ≥19 years should be vaccinated with two doses of recombinant zoster vaccine (RZV) for the prevention of herpes zoster and related complications [48].

## 6. Kidney Transplant Candidates

All-stage chronic kidney disease (CKD) patients should receive all the routinely recommended vaccines according to their age group and associated risk factors [8]. There is no consensus on the ideal CKD stage to administer vaccines, since administration too early might lead to “unnecessary” immunization, while administration in advanced stages may be less immunogenic [9]. As a general recommendation, appropriate complete vaccination should be performed in the pre-kidney transplant (KT) period, at least 4 weeks prior to transplant for LAVs and at least 2 weeks prior to transplant for inactivated vaccines [2,38]. Since LAVs are contraindicated after KT, it is mandatory to assess baseline serologic status for MMR in KT candidates, and in vaccine non-immune patients. If seroconversion does not occur, the dose can be repeated once [2,49]. Table 3 reports the recommendations of the vaccination policies in both CKD and KT recipients.

### 6.1. Influenza

Inactivated influenza vaccine should be administered annually since infection exposes considerable morbidity and mortality for CKD patients [2,3,4,8]. Reported rates of seroconversion in dialysis patients are lower than in healthy controls and ranges from 33 to 80%, but vaccine administration is associated with decreased mortality and hospitalization [8]. Potential ways to increase efficacy include increased doses (especially in >65 years), booster doses and adjuvants [3].

### 6.2. Streptococcus Pneumoniae

Pneumonia in CKD patients, frequently caused by *Streptococcus pneumoniae*, is a common infection with increased morbidity and mortality. Pneumococcal immunization is recommended for such patients [8,50]. The dosing schedule in naïve patients includes the PCV13 which stimulates the production of antibodies with higher affinity and also leads to the formation of memory B cells, followed by the PPSV23, at least 8 weeks later, which induces a T-cell-independent response. Patients who previously received PPSV23, should be given PCV13 at least one year later. A PPSV23 booster should be administered five years after initial PPSV23 in subjects <65 years [51]. Evidence of pneumococcal vaccine efficacy in CKD patients is limited; however, both vaccines are immunogenic in patients on dialysis (PCV13 more than PPSV23), with antibody titres waning over time [3,8,50]. Co-administration with inactivated influenza vaccine may have synergistic positive effects [2].

### 6.3. HBV

HBV infection still represents a risk for haemodialysis patients, who are exposed to blood-borne pathogens, especially in endemic regions. The HBV vaccine therefore represents an important protective measure for both the patients and the staff who cares for them [8]. One of the major challenges is due to a poor serologic response to HBV vaccine in this population, particularly when already on dialysis at the time of vaccination: a seroconversion rate of 44% in case of end-stage renal disease (ESRD) vs. 90% in stage 3/4 CKD vs. 96% in healthy controls is reported [52].

The classical vaccination schedule is based on three doses of recombinant vaccine at time 0, 1 and 6 months. Repeating the full vaccination schedule can be attempted in non-responders, while a booster dose is recommended when the anti-hepatitis B surface antigen (anti-HBs) titre declines below <10 mIU/mL at annual periodic monitoring [2,3,8,52]. Several methods have been evaluated to enhance the vaccine response in CKD patients, such as high dose (40 μg vs. 20 μg), intra-dermal administration, novel epitopes and adjuvant [2,3,52].

### 6.4. SARS-CoV-2

CKD patients, especially those on dialysis, are at increased risk of SARS-CoV-2 infection, hospitalization and mortality [53,54,55,56]. Available data suggest a suboptimal humoral response to vaccines in comparison with the general population, particularly in dialysis patients and with adenoviral vectors, compared with mRNA vaccines [48]. More recent observations suggest a significant increase in antibody levels after a third vaccine dose [51,53,57].

Since a gradual waning of antibody levels has been described over time, the administration of a booster dose of vaccine against COVID-19 4–6 months after the primary series should be encouraged in all patients on dialysis. Further data on long-term outcomes and vaccine efficacy are needed to adopt the best vaccination strategy, since cellular response, required for an optimal clinical protection, seems to also be inferior in the dialysis population compared to controls [58].

### 6.5. HZV

CKD patients have a higher incidence of HZV than the general population, and it has been suggested that HZV infection can promote renal function deterioration [3,8]. The two currently available anti-HZV vaccines, the live attenuated and inactivated recombinant, can both be administered in CKD patients, although the recombinant one is preferred due to its greater efficacy and long-lasting immunity [3,8]. Vaccination within 2 years after starting dialysis was associated with greater protection [3].

### 6.6. Other Vaccinations

Patients submitted to splenectomy, or for whom therapy with eculizumab is expected (before or after transplant), should receive the meningococcal vaccine with two doses of quadrivalent vaccine (against serogroups A, C, Y and W) and two doses of serogroup B vaccine [50]. CKD patients should receive the usual routine immunizations, such as a Td booster every 10 years (or earlier in the setting of wound care) or TDaP if they did not receive a previous vaccine as a child [2]. Existing data suggest a lower tetanus vaccine immunogenicity in case of renal diseases when compared to healthy individuals [59].

Since it has been described that KT boys and girls have a decreased response to the HPV vaccination compared to CKD and dialysis patients, it is important to advocate for HPV immunization prior to KT [60].

In case of travel to at-risk regions, specific vaccinations, i.e., HAV, yellow fever, cholera and typhoid, should be administered according to usual recommendations. Safety and efficacy data are generally not available. If a travel to yellow fever zones is expected after KT, vaccination should be considered at least one month before transplantation [50].

## 7. Kidney Transplant Recipients

An optimal timing for vaccine administration after KT has not been established, but most recommendations agree to wait 3 to 6 months after transplant or treatment for rejection before vaccinating these patients [8,9,50]. LAVs, such as MMR, HZV and yellow fever, are contraindicated in the post-KT period because of the risk of viral replication, while inactivated vaccines have proven to be safe and effective and can be administered without significant risk of rejection [8].

### 7.1. Influenza

Annual administration of one of the inactivated vaccine formulations (quadrivalent or trivalent) is strongly recommended [3]. If a community outbreak of infection occurs before the third month after KT, anticipation of influenza vaccine, as early as one month after transplant should be considered [3,8,9,50,61]. KT recipients treated with a daily dose ≥2 g of MMF and >65 years have shown a reduced humoral response to vaccination [8]. Different approaches to improve response rates have been tested, such as repeating vaccination after 4 to 8 weeks or administering a higher dose of antigen [2,3,8]. The latter strategy has been associated with increased immunogenicity and is generally recommended in patients >65 years [2,3,8,50].

### 7.2. Streptococcus Pneumoniae

Immunization with PCV13 followed by PPSV23 at least 8 weeks later should be given to all KT recipients. Repeating PCV13 administration may be considered in patients >65 years [3,8,9]. Data on the response to PCV13 followed by PPSV23 in KT recipients are not available in the literature [9]. Available heterogeneous data, using various regimens and 7-valent conjugate vaccines, showed a serological response not significantly different from that of the general population, but with lower antibody titres and a faster decline [62].

### 7.3. HBV

Thanks to the policy of universally vaccinating haemodialysis patients, the majority of KT recipients should have been immunized before KT [9]. The immunological status by dosing anti-HBs titres should be periodically assessed after KT, especially in the case of ongoing risk for HBV exposure or traveling to high-risk areas [61]. Revaccination can be considered in non-immune patients or in those with declining immunity (anti-HBs <10 mIU/mL). Response rates after vaccine administration for HBV in the post-transplantation period varies greatly, titres decline more rapidly and booster doses produce suboptimal responses [63].

### 7.4. SARS-CoV-2

Available data suggest a significant lower vaccine response in KT recipients despite additional booster doses, warranting an augmented adherence of patients to protective measures and the need for alternative strategies to prevent severe infection, like the use of monoclonal antibodies or antiviral therapies [64].

### 7.5. HZV

KT recipients are exposed to a high risk of HZV infection and related complications. The recombinant adjuvanted inactivated varicella zoster vaccine should be given to all KT recipients, ideally 6 to 12 months after transplantation [8,9]. Available data show the vaccine to be safe and immunogenic after two doses compared to a placebo [65].

### 7.6. Other Vaccinations

Vaccination against meningococcal disease in KT patients deserves special consideration in those scheduled to receive eculizumab, whether for the prevention of graft rejection or for the treatment of atypical haemolytic-uremic syndrome. The immunization plan should include vaccines against serotype B and against serotypes A, C, Y and W, and optimal timing should be at least two weeks before the initiation of eculizumab therapy. Unfortunately, responses to vaccines are generally poor, even after a repeated dose, and nearly 50% of patients have been shown to develop protective antibodies [8], and antibiotic prophylaxis against Neisseria meningitides is usually provided [8,9,50]. Other safe and recommended vaccination in post-KT (according to previous immunological status) may include HAV, HPV, *tetanus*, *diphtheria* and *pertussis*.

KT recipients should be warned about the potential risks of visiting regions which require special vaccination. LAVs, as with yellow fever, oral *Salmonella typhi* and *Cholera*, are not recommended, while inactivated vaccines, i.e., intramuscular *Salmonella typhi*, *Japanese encephalitis*, tick-borne encephalitis and *Rabies*, are safe and strongly recommended according to regional indications [8,9,50].

In order to ensure an adequate protection from infections, vaccination strategies are also crucial also for close contacts: family members, health care workers and pets. This category of people should receive all the recommended immunizations, especially the annual influenza vaccine and SARS-CoV-2 immunization. Live oral polio is contraindicated in close contacts due to the risk of transmission [9,65]. For most LAVs, no special precautions are required, but some cases do require them. In particular, KT patients should be isolated from vaccine recipients presenting rash, while other precautions involve the risk of virus shedding in the stool for one week after the live attenuated cholera vaccine, and for two to four weeks after rotavirus vaccines [8,9,65].

## 8. Liver Transplant Candidates

Current guidelines recommend that in chronic liver disease, vaccines should be administered as soon as possible due to a higher immunogenicity in a prior phase of the natural course of the disease. For the same reason, it is better to vaccinate patients before liver transplant (LT), prior to administering high levels of IS regimes [66,67]. Particularly, inactivated vaccines should be used at least two weeks prior to IS therapy while live vaccines should be given ≥4 weeks prior to IS therapy and should be avoided within two weeks of the start of these drug regimens [12]. Table 4 reported the vaccine recommendations in chronic liver disease (CLD) patients and in LT recipients.

### 8.1. Influenza

Many studies demonstrate the safety and tolerability of the influenza virus vaccination in cirrhotic patients with the reduction of episodes of decompensation. The dose recommended is one dose of tetravalent inactivated vaccine every year, while the LAV can be administered before LT, with a temporary contraindication to transplant for two weeks after vaccination [68,69].

### 8.2. Streptococcus Pneumoniae

The pneumococcal invasive disease is one of the most important causes of hospitalization and mortality in patients with advanced chronic liver disease. Adults with chronic liver disease and LT candidates should receive a dose of PPSV23 if they have never received a prior dose. When both PPSV23 and PCV13 are indicated, PCV13 should be administered 8 weeks prior to PPSV23. If the patient was already vaccinated for PPV23, the injection of a dose of PCV13 is recommended at least a year after the PPV23 injection and another dose of PPV23 at least 5 years after the first dose. If the patient is already vaccinated with PCV13 and PPV23, one more dose of PPV23 may be administered at least 5 years after the previous one. A small study evaluating the serological response of 45 candidates for LT receiving PPSV23 suggested that this vaccine was not very effective after LT [70].

### 8.3. HBV

Patients with chronic liver disease have a global lower immune response and this has been established for the HBV vaccine too. After the HBV vaccine, the seroconversion rate was 94% vs. 39% in patients with steatosis and cirrhosis, respectively [71]. A prospective study in chronic HCV infected patients undergoing three doses at 0, 30 and 180 days has shown 38% vs. 85% seroprotection rates in CLD or cirrhotic patients compared to healthy subjects [72]. HBV vaccination is recommended in all anti-HB-negative CLD patients. If a post-vaccination anti-HBs concentration of ≥10 mIU/mL is not attained, a second three-dose series of HBV vaccine should be administered using standard or high doses [17].

### 8.4. SARS-CoV2

Patients with advanced liver disease have a high risk of acute decompensation and liver failure with increased mortality because of SARS-CoV-2 infection and its sequelae [73]. For these reasons, even if long-term safety data are not available, EASL guidelines suggest mRNA vaccines (Pfizer-BioNTech^®^ and Moderna^®^ with a schedule of two doses 21 and 28 days apart, respectively) to all patients with advanced cirrhosis, liver decompensation and hepatobiliary cancers [74].

### 8.5. HZV

Two doses of RZV for the prevention of HZV-related complications are recommended for patients ≥19 years who are or will be immunodeficient or immunosuppressed because of therapy [12,48].

### 8.6. Other Vaccinations

In the setting of an advanced chronic liver failure, HAV infection can lead to acute decompensation and increased mortality. For this reason, vaccination is recommended in all cirrhotic patients and, due to a significantly major immune response in the earlier phases of the disease, it should be administered as soon as possible [75].

Data on HPV infections in cirrhotic patients are scarce. The HPV vaccine is a non-infectious recombinant vaccine obtained from purified virus-like particles of the L1 proteins of HPV and it is recommended in the general population before the viral exposure. The immunization protocols in LT candidates should follow the current guidelines:HPV vaccination is recommended at age 11 or 12 years through age 26 years for “naïve” patients;For adults aged 27–45 years, HPV vaccination can be administered on the basis of patient’s benefit, since most people in this age range could have been already exposed to the virus [76];In patients who have already received two doses of HPV at 12 years of age, it is not necessary to obtain other doses, while, if the SOT candidate has never been vaccinated, three doses are recommended (0–2–6 months).

## 9. Liver Transplant Recipients

### 9.1. Influenza

The influenza vaccine in post-LT recipients usually reduces disease severity [77]. The recommended schedule is one dose of the tetravalent-inactivated vaccine once a year, while the LAV should not be used after transplantation [78]. A study comparing the high- and standard dose of influenza vaccines in 172 SOT recipients has shown seroconversion in 79% and 56%, respectively, in SOT recipients [79].

### 9.2. Streptococcus Pneumoniae

If the transplant recipient is naïve for PCV13/PPSV23 vaccines before LT, vaccination should be considered 3 to 6 months after LT. PPSV23 should be administered ≥8 weeks after PCV13 [17].

### 9.3. HBV

The HBV vaccine is recommended in all patients who have not received any previous HBV immunization and the schedule is three high doses at 0, 3 and 6 months, 2 to 6 months after LT [17]. A study conducted in 140 LT recipients reported a 40% response rate to the HBV vaccine with a rapid decline in antibody titres, probably due to immunosuppression anyway [80].

### 9.4. SARS-CoV-2

The current recommendation is to vaccinate as soon as possible before starting IS therapy because the immunogenicity and efficacy could be lower in transplanted patients. A prospective study evaluating 658 SOT recipients receiving two doses of the SARS-CoV-2 mRNA vaccine showed that 15% of them had measurable serum antibody levels after two doses and 46% had no response after dose 1 or dose 2; 39% had no response after dose 1 but subsequently developed measurable serum antibody levels after dose 2 [39]. Another study in 80 LT recipients receiving two doses of the SARS-CoV-2 mRNA vaccine showed that only 47% of LT recipients developed antibodies vs. 100% of healthy controls, and that in LT recipients with a measurable serology, the average SARS-CoV-2 IgG titres were lower compared to the healthy controls: 95 vs. 200 AU/mL, respectively [81]. A study conducted in Germany compared 24 LT recipients to 19 healthy controls who received up to four doses of an mRNA vaccine for SARS-CoV-2 infection. Even if LT recipients had significantly lower levels of spike-specific IgG after three mRNA vaccine doses compared to the control group, most of them showed an overall robust humoral and cellular memory response [82]. Regarding additional booster doses, they need to be administered upon the individual immune response for a possible personal benefit.

### 9.5. HZV

Two doses of RZV for the prevention of HZV-related complications are recommended for patients ≥19 years who are or will be immunodeficient or immunosuppressed because of therapy [17,48].

### 9.6. Other Vaccinations

The antibody response after the HAV vaccination in the post-LT period has shown a seroconversion in 41% of the patients after the primary dose and a response rate of 97% for the patients receiving the secondary dose, in line with healthy controls [83].

In LT recipients, there is an increased risk of HPV-associated cancers of the anogenital area and oropharynx [84]. Due to a consistent increase in cervical neoplasia and invasive cervical cancer, long-term surveillance and treatment for a continued risk long after LT are mandatory, emphasizing the need for screening throughout a woman’s lifetime. The dose recommended for immunocompromised people is a three-dose series at 0, 2 and 6 months.

## 10. Conclusions

Despite the advances in immunosuppression and medical management, infectious diseases are a well-known cause of morbidity and mortality in SOT recipients compared to immunocompetent individuals. Vaccination is the most efficient and cost-effective intervention to prevent infectious diseases; the response to vaccines, however, is highly dependent on a fully functioning immune system.

Patients with organ failure waiting for transplantation are at increased risk of infections and acute decompensation with subsequent high morbidity and mortality. On the other hand, in post-transplant recipients, immunosuppression is the major predisposing factor for the development of infectious diseases, bearing a strong impact on the rate and severity of the infection’s manifestations as well as the patients’ survival, since antimicrobial treatments are usually less effective than in immunocompetent patients.

In order to ensure adequate protection from infections, vaccination strategies are crucial also in close contacts: family members, health care workers, and pets. This category of people should receive all the recommended immunizations, especially annual influenza vaccine preferably with the inactivated influenza vaccine as well as against MMR and varicella, in order to minimize the possibility of exposure to wild-type viruses, and SARS-CoV-2 immunization.

Vaccination represents a major preventive measure against specific infectious risks in these populations, especially from the perspective of a future SOT, when IS therapy will boost the infectious risk and further limit the immune system responsiveness. Although the patients with end-stage organ disease may have a reduced serological response to vaccines due to the compromised immune response compared to healthy controls, vaccines clearly demonstrate superior immunogenicity when administered earlier in the course of the chronic disease. Accordingly, the “ideal” strategy is to timely verify the vaccination status of all the patients with advanced diseases, aiming at promptly identifying and resolving any possible immunity gaps before SOT.

## Figures and Tables

**Table 1 vaccines-11-01611-t001:** Vaccine recommendations in lung transplant (Lu-T) candidates and recipients.

	Pre Transplant		Post Transplant	
	Timing	Schedule	Timing	Schedule
Influenza	Inactivated, at least 2 weeks prior Lu-T	Single dose	Inactivated, 1 month after Lu-T	Single dose
*Streptococcus pneumoniae*	At least 2 weeks prior Lu-T	A single dose of conjugate vaccine and a dose of polysaccharide vaccine at least 8 weeks after; a second dose can be administered after 5 years	3–6 months after Lu-T	Same as pre-Lu-T schedule
HBV	At least 2 weeks prior Lu-T	Three-dose series with the first 2 doses separated by ≥4 weeks, and a third dose after 4–6 months	3–6 months after Lu-T	Same as pre-Lu-T schedule
HAV	At least 2 weeks prior Lu-T	Two-dose series separated by 6–12 months	3–6 months after Lu-T	Same as pre-Lu-T schedule
SARS-CoV-2	At least 2 weeks prior Lu-T	Three-dose series + two boosters’ dose	First vaccination/second dose for at least 1 month after Lu-T and for at least 3 months	Same as pre-Lu-T schedule
HZV	At least 2 weeks pre-Lu-T	RZV 2 doses, spaced 2 to 6 months apart	Not recommended	Same as pre-Lu-T schedule
Meningococcal	At least 2 weeks prior Lu-T	Single dose	3–6 months after LT	Same as pre-Lu-T schedule
HPV	At least 2 weeks pre-Lu-T	Three-dose series at 0–2–6 months	3–6 months after Lu-T	Same as pre-Lu-T schedule
MMR	At least 4 weeks pre-Lu-T	Two-dose series, at least 4 weeks apart. Should be completed at least 2 weeks before Lu-T	Not recommended	-
Tetanus	At least 2 weeks pre-Lu-T	Repeat vaccination every 10 years	3–6 months after Lu-T	Same as pre-Lu-T schedule
Tdap/Td	At least 2 weeks pre-Lu-T	A single dose, repeat vaccination every 10 years	3–6 months after Lu-T	Same as pre-Lu-T schedule
*Haemophilus influenza* type B	At least 2 weeks pre-Lu-T	Single dose	3–6 months after Lu-T	Same as pre-Lu-T schedule
Rabies	At least 2 weeks pre-Lu-T	Three-dose series at 0–1–12 months	3–6 months after Lu-T	Same as pre-Lu-T schedule
BCG	Not recommended	-	Not recommended	-
Yellow fever	At least 4 weeks pre-Lu-T	A single dose, at least 10 days before entering an endemic area	Not recommended	-
Polio	At least 2 weeks pre-Lu-T	Three-dose series with the first 2 doses separated by 4–8 weeks, and a third dose after 6–12 months	3–6 months after Lu-T	Same as pre-Lu-T schedule
Smallpox	At least 4 weeks pre-LT	Single dose	Not recommended	-
Cholera	At least 2 weeks pre-LT	Two doses 1–6 weeks apart; the course should be completed at least 1 week before any exposure to cholera; for continued protection, a single booster dose within 2 years is recommended	not recommended	-

LAV, live attenuated vaccines; LZV, live attenuated zoster vaccine; PCV13, 13-valent pneumococcal conjugate vaccine; PPSV23, 23-valent pneumococcal polysaccharide vaccine; RZV, recombinant zoster vaccine; HAV, hepatitis A virus; HBV, hepatitis B; HPV, human papilloma virus; HZV, herpes zoster virus; MMR, measles, mumps and rubella; Tdap, tetanus toxoid, reduced diphtheria toxoid, acellular pertussis; Td, tetanus diphtheria; BCG, bacille Calmette–Guerin.

**Table 2 vaccines-11-01611-t002:** Vaccines recommended in heart transplant (HT) candidates and recipients.

	Pre-Transplant		Post-Transplant	
	Timing	Schedule	Timing	Schedule
Influenza	Each winter	Single dose	3–6 months after HT	Single dose
*Streptococcus pneumoniae*	PCV e PPV prior HT	1–8 weeks	3–6 months after HT	1–8 weeks
HBV	Two doses, 1 month prior HT	0–1–6 months	3–6 months after HT	Last doses also after HT
HAV	One dose prior HT	0–6 months	3–6 months after HT	Second dose also after HT
SARS-CoV-2	Two doses, 1 month prior HT	0–1 month	3–6 months after HT	Three doses also after HT
HZV	Completed 1 month prior HT	Two doses 0–2 months	3–6 months after HT	Two doses 0–2 months
HPV ^§^	-	0–2–6 months	3–6 months after HT	Last dose also after HT
Tdap/Td, *Haemophilus influenzae* type B	At least 2 weeks prior HT	As general population	3–6 months after HT	As general population
Rabies	At least 2 weeks prior HT	As general population	3–6 months after HT	0, 7, 21 days
Tetanus	1 month prior HT	If never vaccinated 0–2–6/12 months, otherwise 1 booster dose	3–6 months after HT	-

HAV, hepatitis A virus; HBV, hepatitis B; HPV, human papilloma virus; HZV, herpes zoster virus; MMR, measles mumps and rubella; TDap, tetanus toxoid, reduced diphtheria toxoid, acellular pertussis; Td, tetanus diphtheria; BCG, bacille Calmette–Guerin. ^§^ <50 years, MMR in non-immune pre-HT patients but not recommended in post-HT patients. Yellow fever, Polio, type B rotavirus, BCG and cholera are not recommended in HT candidates and recipients.

**Table 3 vaccines-11-01611-t003:** Vaccines recommendations in chronic kidney disease (CKD) patients and kidney transplant (KT) recipients.

	Pre-Transplant		Post-Transplant	
	Timing	Schedule	Timing	Schedule
Influenza	Each winter, at least 2 weeks prior KT	Single dose	3–6 months after KT (as early as 1 month after KT in case of outbreak but avoiding LAV)	Single dose
*Streptococcus pneumoniae*	At least 2 weeks prior KT	PCV13 followed by PPSV23 8 weeks later	3–6 months after KT	PCV13 followed by PPSV23 8 weeks later
HBV	At least 2 weeks prior KT	0, 1 and 2 and/or 6 months depending on type of vaccine	3–6 months after KT	0, 1 and 6 months
HAV	At least 2 weeks prior KT	As general population	3–6 months after KT	As general population
SARS-CoV-2	At least 2 weeks prior KT	As general population	3–6 months after KT	As general population
HZV	2 weeks for RZV	As general population	HZV not recommended, RZV, 3–6 months after KT	2 doses at least 8 weeks apart
Meningococcal	At least 2 weeks prior KT	As general population	3–6 months after KT	As general population
HPV	At least 2 weeks prior KT	As general population	3–6 months after KT	0, 2, 6 months, monovalent vaccine up to 45 years
MMR	At least 4 weeks pre-KT	As general population	Not recommended	-
Tetanus, Tdap/Td, Haemophilus influenza type B	At least 2 weeks prior KT	As general population	3–6 months after KT	As general population
Yellow fever	At least 4 weeks pre-KT	As general population, in case of travel to at-risk areas	Not recommended	-
Polio	At least 4 weeks pre-KT	As general population	Inactivated, 3–6 months after KT	As general population
Rabies	At least 2 weeks prior KT	As general population	3–6 months after KT	0, 7, 21 days
Rotavirus, BCG, Smallpox, Cholera	At least 4 weeks pre-KT	As general population	Not recommended	-

KT, kidney transplant; CKD, chronic kidney disease; LAVs, live attenuated vaccines; LZV, live attenuated zoster vaccine; PCV13, 13-valent pneumococcal conjugate vaccine; PPSV23, 23-valent pneumococcal polysaccharide vaccine; RZV, recombinant zoster vaccine; HAV, hepatitis A virus; HBV, hepatitis B; HPV, human papilloma virus; HZV, herpes zoster virus; MMR, measles, mumps and rubella; Tdap, tetanus toxoid, reduced diphtheria toxoid, acellular pertussis; Td, tetanus diphtheria; BCG, bacille Calmette–Guerin.

**Table 4 vaccines-11-01611-t004:** Vaccines recommendations in chronic liver disease and in liver transplant (LT) recipients.

	Pre-Transplant		Post-Transplant	
	Timing	Schedule	Timing	Schedule
Influenza	Inactivated, at least 2 weeks prior LT	Single dose	Inactivated, 1 month after LT	Single dose
*Streptococcus pneumoniae*	At least 2 weeks prior LT	Not previously vaccinated: one dose of PCV 13 followed at least 8 weeks by one dose of PPV23. Vaccinated with PPV23: one dose of PCV 13 at least 1 year after PPV23 and one dose of PPV23 at least 5 years after the first dose	3–6 months after LT	PCV13 followed by PPSV23 8 weeks later
HBV	At least 2 weeks prior LT	0, 1 and 6 months	3–6 months after LT	0, 1 and 6 months
HAV	At least 2 weeks prior LT	Two doses six months apart or if combined with HBV vaccine 3 doses over a six months period	3–6 months after LT	Two doses six months apart or if combined with HBV vaccine 3 doses over a six months period
SARS-CoV-2	At least 2 weeks prior LT	As general population	3–6 months after KT	As general population
HZV	At least 4 weeks pre-LT	Two LAV doses 4 weeks apart for varicella vaccine; one dose for zoster in HZV IgG positive candidates	Not recommended	-
Meningococcal, HPV	At least 2 weeks prior LT	As general population	3–6 months after LT	As general population
MMR	At least 4 weeks pre-LT	As general population	3–6 months after LT	As general population
Tetanus, Tdap/Td, *Haemophilus influenza* type B, Rabies	At least 2 weeks prior LT	As general population	3–6 months after LT	As general population
Yellow fever	At least 4 weeks pre-LT	As general population	Not recommended	-
Polio	At least 4 weeks pre-LT	As general population	Inactivated, 3–6 months after LT	As general population
Rotavirus, BCG, Smallpox, Cholera	At least 4 weeks pre-LT	As general population	Not recommended	-

LAVs, live attenuated vaccines; LZV, live attenuated zoster vaccine; PCV13, 13-valent pneumococcal conjugate vaccine; PPSV23, 23-valent pneumococcal polysaccharide vaccine; RZV, recombinant zoster vaccine; HAV, hepatitis A virus; HBV, hepatitis B; HPV, human papilloma virus; HZV, herpes zoster virus; MMR, measles mumps and rubella; Tdap, tetanus toxoid, reduced diphtheria toxoid, acellular pertussis; Td, tetanus diphtheria; BCG, bacille Calmette–Guerin.

## Data Availability

Not applicable.

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
