# Peer review of "Vaccination Recommendations in Solid Organ Transplant Adult Candidates and Recipients"

_vaccines, 2023, doi:10.3390/vaccines11101611_

Round 1

Reviewer 1 Report

Interesting laundry list of vaccinations in patient with organ failure and in transplant recipients.

The English is good but not perfect, and needs to be fine-tuned by a native English writer.

Live virus vaccines have been given to pediatric organ transplant recipients, with good safety and efficacy, and this should be discussed by the authors.

Additionally:

1. What is the main question addressed by the research?--A REVIEW OF VACCINATION IN SOLID ORGAN TRANSPLANTATION

2. Do you consider the topic original or relevant in the field? Does it address a specific gap in the field?--NOT ESPECIALLY ORIGINAL

3. What does it add to the subject area compared with other published material?--NOT THAT MUCH

4. What specific improvements should the authors consider regarding the methodology? What further controls should be considered?--THEY SHOULD DISCUSS THE FACT THAT PEDIATRIC SOLID ORGAN TRANSPLANT PATIENTS HAVE RECEIVED LIVE VIRUS VACCINES POST TRANSPLANT WITH GOOD SAFETY AND EFFICACY.

5. Are the conclusions consistent with the evidence and arguments presented and do they address the main question posed?--REALLY MORE OF A LAUNDRY LIST PAPER

6. Are the references appropriate?--NO; THEY NEED TO CITE THE PEDS PAPERS

The English is good but not perfect, and needs to be fine-tuned by a native English writer.

Author Response

Reviewer 1

1) The English is good but not perfect, and needs to be fine-tuned by a native English writer.

We have submitted the paper for an extensive revision by a native English writer.

2) Live virus vaccines have been given to paediatric organ transplant recipients, with good safety and efficacy, and this should be discussed by the authors.

We thank the reviewer for the suggestion. Although live virus vaccines have been given to paediatric organ transplant recipients, with good safety and efficacy, there are scattered reports of such policy in the adult population. Live vaccines are generally to be avoided in recipients of SOT; this view has been recently questioned. In particular, a 2-day consortium of experts in infectious diseases, transplantation, vaccinology, and immunology has recently published expert recommendations on the topic Notably, the risk of live vaccines in SOT recipients is linked to vaccine-mediated disease. In fact, the use of Yellow Fever Vaccine has been associated to acute antibody-mediate renal graft rejection. In some cases, if a corresponding inactivated vaccine is available, this is the one to be employed (e.g., in case of typhoid vaccination). Unfortunately, no data exist concerning the protection afforded by recombinant zoster vaccines, against a primary VZV infection. The above notwithstanding, all efforts should be pursued, in order to vaccinate with live vaccines (principally MPR and varicella) the family members and the closest relatives of immunosuppressed patients ("cocooning" vaccination strategy). (See introduction page 4 and 5)

3) Do you consider the topic original or relevant in the field? Does it address a specific gap in the field? --NOT ESPECIALLY ORIGINAL

The aim of our review is to assess the main indications to active immunization in adult patients with advanced heart, kidney, liver and lung disease and in SOT recipients. A comprehensive review manuscript including all the SOT seem to be “handy” and rather unprecedent.

4) What does it add to the subject area compared with other published material? --NOT THAT MUCH

We believe that our manuscript has the merit of being a comprehensive review of the existing material which is generally published in field-specific journals

5) Are the conclusions consistent with the evidence and arguments presented and do they address the main question posed? --REALLY MORE OF A LAUNDRY LIST PAPER

The conclusions are consistent with the current evidence

6) Are the references appropriate? --NO; THEY NEED TO CITE THE PEDS PAPERS

The aim of our review is to assess all the main indications to active immunization in adult SOT recipients. However, according to reviewer suggestion we have added one reference related to the topic of live virus vaccines in paediatric patients and the argument is discussed in the manuscript

Reviewer 2 Report

The manuscript by Viganò et al describes the vaccination strategies and Policies in Solid Organ Transplant adult candidates and recipients. The manuscripts explains the importance of vaccination in prevention   against specific infectious risks in this population.

The following are a few suggestions, which may be addressed before considering the manuscript for publication.

  1. Although vaccination can be considered the best preventive strategy against infection among solid organ transplant recipients, the extent of the immune response and protective efficacy of the vaccines in these subjects is the major bottleneck besides the adverse events associated with the vaccine, which may be accentuated in such population. Therefore, if authors can do a meta-analysis of such data from different studies it would significantly improve the manuscript.
  2. The authors have described the vaccination policies in the SOT cases of heart, lung, liver and kidney. However, pancreas and intestine SOT may also be included in the scope of the review.
  3. Manuscript needs to be checked thoroughly for grammatical errors.
  4. Tables can be removed from the manuscript and can be presented as supplementary data.

Good

Author Response

1) Although vaccination can be considered the best preventive strategy against infection among solid organ transplant recipients, the extent of the immune response and protective efficacy of the vaccines in these subjects is the major bottleneck besides the adverse events associated with the vaccine, which may be accentuated in such population. Therefore, if authors can do a meta-analysis of such data from different studies it would significantly improve the manuscript.

We wish to thank the reviewer for the suggestion. On one hand we acknowledge that a meta-analysis of all the available data on this topic might be of interest, however, it goes beyond the aims of our manuscript and it would certainly require a much longer time for the preparation. We believe that a dedicated metanalysis might be the core of a different publication.

2) The authors have described the vaccination policies in the SOT cases of heart, lung, liver and kidney. However, pancreas and intestine SOT may also be included in the scope of the review.

We wish to thank the reviewer for the advice of including both small bowel and pancreas transplantation to the manuscript. We agree upon the fact that they might deserve a specific chapter despite the rarity of both types of transplants. However, available data in this setting are truly scattered. On the other hand, since in most cases the intestinal transplant is performed in the context of a multivisceral procedure with the liver and the pancreas transplant is associated with the kidney, we have included the recommendations in these settings as inferences from other combined organs. Therefore, we have dealt with these two transplants similarly to liver and kidney recipients, based on the respective immunosuppression regimens. We have also added a note in the text (See page 5)

3) Manuscript needs to be checked thoroughly for grammatical errors.

We have submitted the paper for an extensive English revision, by a native speaking writer.

4) Tables can be removed from the manuscript and can be presented as supplementary data.

We believe that the tables are easy to read and quick to consult. However, we leave this choice to the Editor

Reviewer 3 Report

This is an excellent review article that covers the broad vaccination strategy adopted for vaccination in solid organ transplant recipients. It is more or less similar across the world. However, data are presented mainly from Western countries, where the prevalence of most of the infections is not very frequent, like African and Asian countries, where due to the high infection burden, the immunogenic response remains relatively higher against these common infections and even people have pre-formed memory cell and even vaccination boost the seroconversion response many folds despite of immunosuppression. i.e. patients with liver cirrhosis and renal transplant recipients had higher seroconversion rates before anti-SARS-CoV-2 vaccination in the Indian population. (PMID: 36298558) (PMID:35335017), (PMID:36366346).

Further, most of the transplantation that happens in Western countries are cadaveric related, which requires a higher dose of immunosuppression to maintain the optimal immunosuppression in contrast to live-related transplantation, which may be the other cause for a lower seroconversion rate after vaccination.

Putting data from the Asian, African and other regions will further strengthen the study and broader acceptance in the transplant community.    

English is well-written and understandable but needs minimal editing. 

Author Response

1) This is an excellent review article that covers the broad vaccination strategy adopted for vaccination in solid organ transplant recipients. It is more or less similar across the world. However, data are presented mainly from Western countries, where the prevalence of most of the infections is not very frequent, like African and Asian countries, where due to the high infection burden, the immunogenic response remains relatively higher against these common infections and even people have pre-formed memory cell and even vaccination boost the seroconversion response many folds despite of immunosuppression. i.e., patients with liver cirrhosis and renal transplant recipients had higher seroconversion rates before anti-SARS-CoV-2 vaccination in the Indian population. (PMID: 36298558) (PMID:35335017), (PMID:36366346).

We wish to thank the reviewer for the suggestion. However, we truly believe that the recommendations included in this manuscript should aim to homogenize the approach on the principal issues across the world, avoiding to focus on country-specifics differences.

2) Further, most of the transplantation that happens in Western countries are cadaveric related, which requires a higher dose of immunosuppression to maintain the optimal immunosuppression in contrast to live-related transplantation, which may be the other cause for a lower seroconversion rate after vaccination.

We believe that the major principles of immunosuppression are similar between cadaveric or living related donors and therefore they can be assimilated in the recommendations

Reviewer 4 Report

I considered the manuscript entitled “Vaccinaton Policies in Solid Organ Transplant Adult Candidates and Recipients” by Mauro Viganò, et al, that is intended to be published in Vaccines journal.

I enjoyed the manuscript. It deals with a comprehensive revision on vaccination in SOT patients. However, there are some differences when looking to the different organ described. The most complete is the lung transplantation with better wording, and more extensive and detailed description. It should appear as the first one, before cardiac transplantation.

The manuscript is informative and actualized for the young medical community consultation and learning.

Author Response

Reviewer 4

I enjoyed the manuscript. It deals with a comprehensive revision on vaccination in SOT patients. However, there are some differences when looking to the different organ described. The most complete is the lung transplantation with better wording, and more extensive and detailed description. It should appear as the first one, before cardiac transplantation. The manuscript is informative and actualized for the young medical community consultation and learning.

According to reviewer suggestion lung transplantation appears now as the first, before cardiac transplantation.